# What Makes Good Data for Alignment? A Comprehensive Study of Automatic Data Selection in Instruction Tuning

**Wei Liu**[*1]   **Weihao Zeng**[*2]   **Keqing He**[3]   **Yong Jiang**[4]   **Junxian He**[5]

[1]ShanghaiTech University   [2]Beijing University of Posts and Telecommunications
[3]Meituan   [4]Alibaba Group   [5]The Hong Kong University of Science and Technology
`liuwei4@shanghaitech.edu.cn`   `zengwh@bupt.edu.cn`
`junxianh@cse.ust.hk`

## Abstract

Instruction tuning is a standard technique employed to align large language models to end tasks and user preferences after the initial pretraining phase. Recent research indicates the critical role of data engineering in instruction tuning – when appropriately selected, only limited data is necessary to achieve superior performance. However, we still lack a principled understanding of what makes good instruction tuning data for alignment, and how we should select data automatically and effectively. In this work, we delve deeply into automatic data selection strategies for alignment. We start with controlled studies to measure data across three dimensions: complexity, quality, and diversity, along which we examine existing methods and introduce novel techniques for enhanced data measurement. Subsequently, we propose a simple strategy to select data samples based on the measurement. We present DEITA (short for *Data-Efficient Instruction Tuning for Alignment*), a series of models fine-tuned from LLaMA and Mistral models using data samples automatically selected with our proposed approach. Empirically, DEITA performs better or on par with the state-of-the-art open-source alignment models with only 6K SFT training data samples – over 10x less than the data used in the baselines. When further trained with direct preference optimization (DPO), DEITA-Mistral-7B + DPO trained with 6K SFT and 10K DPO samples achieve 7.55 MT-Bench and 90.06% AlpacaEval scores. We anticipate this work to provide tools on automatic data selection, facilitating data-efficient alignment. We release our models as well as the selected datasets for future researches to effectively align models more efficiently.[1]

## 1 Introduction

In the development of large language models (LLMs), aligning LLMs with human preferences is a necessary step for the models to accurately understand human instructions and generate relevant responses. Standard approaches for LLM alignment encompass instruction tuning and reinforcement learning with human feedback (RLHF) (Ouyang et al., 2022). Instruction tuning, or supervised fine-tuning (SFT), refines the pre-trained model using annotated instructional data, often serving as the foundational step before RLHF to facilitate the model's initial alignment (Touvron et al., 2023b). RLHF, on the other hand, leverages reinforcement learning to train models based on annotated feedback on their generated responses. While RLHF has underpinned the development of ChatGPT (OpenAI, 2022) and GPT-4 (OpenAI, 2023), recent researches suggest that instruction tuning in isolation can offer competitive results (Sun et al., 2024; Zhou et al., 2023).

Differing from traditional task-specific fine-tuning where data quantity is paramount, past studies argue that almost all knowledge in LLMs is acquired during pretraining, and instruction tuning is to

---

[*]Equal Contribution. Order determined by random dice rolling. Work done during WL and WZ's visit to HKUST.

[1]Model checkpoints and data resources are available at https://github.com/hkust-nlp/deita.

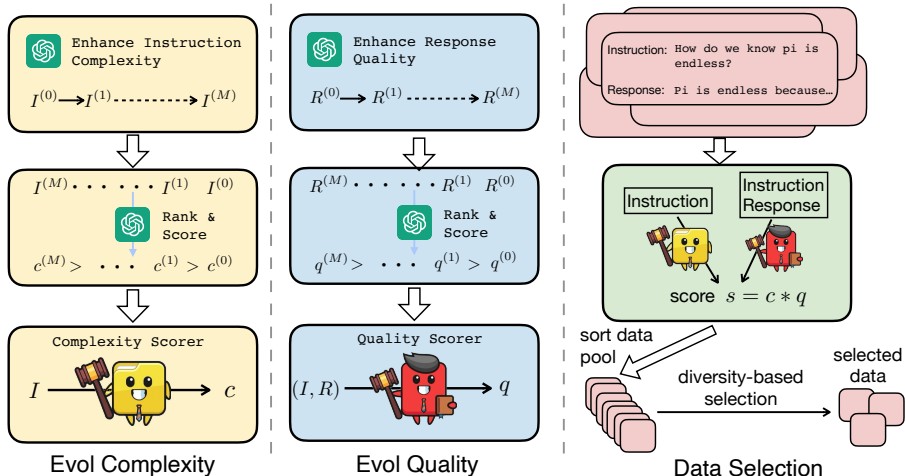

Figure 1: Illustration of the data selection approach. We measure data from three dimensions: complexity, quality, and diversity. $I$ and $R$ represent instruction and response respectively. For EVOL COMPLEXITY and EVOL QUALITY, we first collect samples with varying complexities or qualities through adopting an evolution-based approach following Xu et al. (2023), then we ask ChatGPT (The term "ChatGPT" in this paper refers to gpt-3.5-turbo-0613) to rank and score the variants of the same data sample for a small seed dataset, and we train our own complexity and quality scorers based on these scores. In the last step, we utilize the trained scorers and adopt a score-first, diversity-aware approach to select the "good" data samples, as we detail in §3.

align existing model abilities towards a desired direction (Zhou et al., 2023). As such, a relatively small high-quality dataset has been shown to be sufficient to align LLMs well (Taori et al., 2023; Wang et al., 2023; Zhou et al., 2023; Lu et al., 2023), with dataset sizes spanning from hundreds of thousands to a mere 1000 examples. However, the construction of these datasets in earlier research predominantly relies on heuristic automation (e.g. distillation from ChatGPT) or manual selection, and it remains unclear what defines good data examples for instruction tuning, and how to systematically curate an effective dataset that ensures competitive performance with the least amount of data.

In this work, we seek to define the characteristics of "good data" for instruction tuning, based on which we aim to push *data efficiency* of instruction tuning further in an automatic manner. To this end, we first explore various methods to quantitatively assess data examples from three key dimensions: *complexity*, *quality*, and *diversity*. Our hypothesis posits that the most effective datasets for instruction tuning are those that are complex, of high quality, and diverse. We examine a wide range of data measurement baselines and propose new metrics, which show a stronger correlation with alignment performance after instruction tuning. Drawing inspiration from Xu et al. (2023), we introduce EVOL COMPLEXITY and EVOL QUALITY, which evolve a single data point to produce a series of examples varying in complexity or quality, then we rank and score a small subset of them with ChatGPT and train a scorer to predict these scores, as illustrated in Figure 1. Such an evolution-based method automatically yields examples sorted based on complexity or quality, enabling enhanced and finer-grained data measurements. Combined with our diversity metric via distance of model embeddings, we design a simple strategy to select the most effective data examples from a large data pool. Being simple and effective, our data selection strategy paves the way for a paradigm of data-efficient instruction tuning – where fewer training samples can be automatically selected out to yield performance on par with, or even surpassing, models trained on significantly larger datasets.

We present DEITA (short for Data-Efficient Instruction Tuning for Alignment), a family of models fine-tuned from the LLaMA (Touvron et al., 2023a;b) and Mistral (Jiang et al., 2023) using our proposed technique to maximize data efficiency. On experiments in terms of MT-bench (Zheng et al., 2023), AlpacaEval (Li et al., 2023c), and the Open LLM Leaderboard (Beeching et al., 2023) demonstrate that DEITA is able to outperform or be on par with state-of-the-art instruction following models such as Zephyr (Tunstall et al., 2023), Vicuna (Chiang et al., 2023) and WizardLM (Xu et al., 2023), while using over 10x fewer automatically selected data examples. For example, DEITA-Mistral-7B – our model based on Mistral-7B – achieves 7.22 MT-bench and 80.78% AlpacaEval when trained on only 6K data samples with vanilla SFT training. After equipped with direct preference optimization (DPO, Rafailov et al. (2023)), DEITA-Mistral-7B + DPO, trained with 6K SFT and 10K DPO samples,

obtains 7.55 MT-Bench and 90.06% AlpacaEval. In addition to the DEITA model checkpoints, we release our light yet effective SFT datasets to facilitate alignment for future researches.

## 2 WHAT MAKES GOOD DATA FOR ALIGNMENT?

In this section, we present a comprehensive study of the characteristics of "good data" in instruction tuning. We start by defining the data selection problem (§2.1) under which we analyze different characteristics of data. Then we introduce the experiment setup (§2.2), explore various metrics to assess data and examine their effects in instruction tuning (§2.3 - 2.5).

### 2.1 THE DATA SELECTION PROBLEM

To investigate the characteristics of optimal data for alignment, we analyze within the framework of data selection. In this context, we delve into various metrics to evaluate data, and then employ these metrics to select a subset of data samples for instruction tuning. The resulting alignment performance serves as an indicator of whether the target metric can identify effective instruction tuning data examples. Formally, given a large instruction tuning data pool, $X = \{x_1, x_2, \cdots, x_n\}$, where $x_i$ represents an individual data sample in the form of an instruction-response pair. We aim to select a subset $S_\pi^{(m)}$ of size $m$ from $X$, using a selection strategy denoted by $\pi$. $m$ is the subset size and correlates proportionally with the computation consumed in instruction tuning, thus we also refer $m$ as the *data budget*. Typically, we define a metric to assess data and select data samples based on the metric. Denote the alignment performance after instruction-tuning as $Q$, the optimal data selection strategy $\pi^*$ with data budget $m$ satisfies:

$$\pi^* = \arg\max_\pi Q(S_\pi^{(m)}). \tag{1}$$

In our empirical study following, we will explore a diverse range of data evaluation metrics and their corresponding data selection strategies, by selecting $S_\pi^{(m)}$ according to certain metrics and performing instruction tuning on it. Next, we detail our experimental setup.

### 2.2 EXPERIMENTAL SETUP

In this section, we perform controlled studies on a single metric to evaluate data at a time, and we follow the procedure as (1) selecting a subset $S_\pi^{(m)}$ from a data pool based on a given metric, (2) instruction-tuning a pre-trained model using $S_\pi^{(m)}$, and (3) evaluate the instruction-following abilities of the obtained models. Given a data measurement metric, in this paper, we keep our selection algorithm as simple as possible to maintain its practicality (e.g., selecting the examples with the largest complexity scores), while we leave more advanced selection algorithms as future work.

**Data Pools:** To investigate data selection from large data pools, we construct two data pools with distinct properties to mimic different practical settings: (1) $X_{sota}$, which is constructed by ensembling the training datasets of the state-of-the-art aligned LLMs. This represents the setting where a data pool that is relatively complex, diverse, and of high-quality is available, and we aim to further improve the data efficiency in this case. Specifically, we follow Lu et al. (2023) to combine the datasets WizardLM (Alpaca), WizardLM (ShareGPT), UltraChat (Ding et al., 2023), and ShareGPT (Chiang et al., 2023), resulting in a dataset of 300K samples; and (2) $X_{base}$, which mimics the scenario where the available data pool is overall lower-quality and redundant. This may be closer to the actual setting that people would encounter in practice. For $X_{base}$, we have utilized data from Alpaca (Taori et al., 2023), Dolly (Conover et al., 2023), OAssit (Köpf et al., 2023), and FLAN 2022 (Longpre et al., 2023) to construct a dataset of 100K samples. Such quality-based separation of these open-source instruction tuning datasets roughly aligns with the analysis results in Lu et al. (2023); Li et al. (2023b). Statistics of the two data pools are summarized in Table 1.

**Training and Evaluation:** In the study of this section, we fine-tune the LLaMA-1 13B on the instruction tuning dataset unless otherwise specified. We assume a data budget $m$ of 6K. We use the same hyperparameters for all as detailed in Appendix A. To evaluate alignment performance, we utilize MT-Bench, a challenging benchmark that is commonly adopted to assess the instruction-following ability. MT-bench consists of multi-turn conversations across various domains such as

| Data Pool | Dataset Source | Sample Size |
|---|---|---|
| | ShareGPT | 58 K |
| $X_{sota}$ | UltraChat | 105 K |
| | WizardLM | 143 K |
| | Alpaca | 52 K |
| | Dolly | 15 K |
| $X_{base}$ | OAssit | 10 K |
| | FLAN 2022 | 23 K |

Table 1: Statistics of data pools $X_{sota}$ and $X_{base}$. The Dataset Source indicates the source of the data used for sampling. The Sample Size represents the number of samples in the respective dataset.

| Model | $X_{sota}$ | $X_{base}$ |
|---|---|---|
| Random Selection | 5.84 | 4.93 |
| Instruction Length | 5.89 | 4.00 |
| Perplexity | 4.06 | 1.89 |
| IFD | 5.91 | 2.46 |
| Instag Complexity | 6.18 | 4.98 |
| Direct Scoring (Pool=50K) | 5.16 | 4.87 |
| Instruction Node (Pool=50K) | 5.65 | 4.82 |
| EVOL COMPLEXITY (Pool=50K) | 5.73 | 5.29 |
| EVOL COMPLEXITY | **6.27** | **5.57** |

Table 2: MT-bench of different complexity metrics. All methods select 6K samples. "Pool=50K" denotes the data selection is conducted in a 50K-sized subset due to the cost of using ChatGPT to annotate the entire pool. We include the results of our method on the 50K data pool to make a fair comparison with baselines.

writing, reasoning, math, and coding. GPT-4 is employed as the judge to score model responses in MT-bench, which was found to produce a high agreement with humans (Zheng et al., 2023).

### 2.3 FROM THE COMPLEXITY PERSPECTIVE – EVOL COMPLEXITY

It is often believed that long, difficult, and complex data samples are more beneficial for instruction tuning (Zhao et al., 2023; Cao et al., 2023). For example, Xu et al. (2023) prompt ChatGPT to "evolve" data samples to deliberately increase their complexities, which leads to the state-of-the-art open-source alignment models WizardLM. In this section, we systematically study various metrics to assess the complexity of data and aim to identify the notion of complexity that contributes the most to the instruction-following ability. Concretely, we only consider the complexity dimension and define the selection strategy $\pi_{complexity}$ as selecting the $m$ examples with the highest complexity scores.

**Baselines:** We study several existing methods as our baselines for complexity metrics: (1) *Random Selection* selects examples randomly; (2) *Instruction Length* uses length of instructions as the metric for complexity; (3) *Perplexity* of the responses computed with the pre-trained model in a zero-shot manner is used as the metric, and a large perplexity score typically entails difficulty of the data sample; (4) *Direct Scoring* (Chen et al., 2023) directly prompts ChatGPT to score the difficulty and complexity of instructions; (5) *Instruction Node* (Zhao et al., 2023) uses ChatGPT to transform instructions into a semantic tree and then adopts the number of nodes in the tree as the complexity measure; (6) *Instag Complexity* (Lu et al., 2023) first utilizes ChatGPT to tag the samples based on semantics and intentions, then trains a LLaMA-based tagger on the ChatGPT tags to tag data. They use the number of tags as a proxy for complexity. We adopt their public tagger[2] that is a LLaMA-2 7B model to tag the data. (7) *IFD* (Li et al., 2023a) is a new complexity metric computed based on the response loss. We note that Direct Scoring and Instruction Node are not scalable since they require ChatGPT to annotate the entire data pool, which is expensive. To save cost, we randomly sample 50K examples first from each pool and apply these two baselines. The respective prompts of these baselines (if applicable) are shown in Appendix E.1.

**Evol Complexity:** Taking inspiration from Evol-Instruct which utilizes ChatGPT to evolve examples to become more complex, we propose EVOL COMPLEXITY, a complexity measure based on evolution. Specifically, we collect a small-scale seed dataset, $D = \{(I_1^{(0)}, R_1^{(0)}), \cdots, (I_N^{(0)}, R_N^{(0)})\}$, where $(I_k^{(0)}, R_k^{(0)})$ represents an instruction-response pair. For each instruction sample $I_k^{(0)}$, we use the In-Depth Evolving Prompt (please refer to Appendix E.2 for the prompt) from Xu et al. (2023) to enhance the complexity through techniques such as adding constraints, deepening, concretizing and increasing reasoning steps. After $M$ iterations, we obtain a set of instructions across different complexities for $I_k$, $\{I_k^{(0)}, \cdots, I_k^{(M)}\}$. Here we set $M$ as 5 to obtain 6 variations in total.

As illustrated in the left part of Figure 1, we then ask ChatGPT to rank and score these 6 samples (prompt in Appendix E.2), obtaining the complexity scores $c$ corresponding to the instructions. We emphasize that, distinct from direct scoring, we give ChatGPT all 6 samples within one prompt – these

---

[2]https://github.com/OFA-Sys/InsTag

samples represent different evolution stages of the same original sample and such a scoring scheme helps ChatGPT capture the small complexity differences among them, which leads to complexity scores to achieve finer-grained complexity differentiation among samples. We find that this is critical since otherwise, ChatGPT tends to assign similar scores to most examples, as we showcase in Appendix B. After obtaining ChatGPT scores on the small seed dataset, we use the scores to train an LLaMA-1 7B model to predict the complexity score given the input instruction. In multi-turn dialogues, we score each turn separately and use the sum of them as the final score. Across this paper, we use 2K examples randomly sampled from the Alpaca (Taori et al., 2023) as the seed dataset.

**Results:**   Table 2 presents the results of selecting the 6K data samples from $X_{sota}$ and $X_{base}$ using various complexity metrics. Our Evol-Complexity leads to the best alignment performance. We observe that while Instag Complexity performs well on $X_{sota}$, its performance on $X_{base}$ is only slightly better than Random Selection. In contrast, EVOL COMPLEXITY achieves superior performance on both datasets, indicating strong robustness across different dataset pools. Our method also has significant advantages compared to methods that heavily rely on annotating with ChatGPT, such as Direct Scoring and Instruction Node. The results also suggest that instruction lengths are not good indicators of preferred data by alignment. Interestingly, perplexity, an intuitive measure for complexity, produces much worse results than the random selection baseline. Through further investigation, we found that samples with large perplexity typically exhibit very short responses.

## 2.4   FROM THE QUALITY PERSPECTIVE – EVOL QUALITY

Generally, LLMs that deliver accurate, detailed, and helpful responses are favored by people, as indicated by Zheng et al. (2023). In this section, we carry out a controlled study to examine various metrics used for evaluating the *quality* of samples. Drawing parallels with EVOL COMPLEXITY, we devise a selecting strategy $\pi_{quality}$ based on quality, enabling us to select $m$ samples with the highest quality scores according to different measurements. Following this, we will introduce the examined baseline and our newly proposed methods for assessing quality.

**Baselines:**   We examine the existing methods that serve as baselines for quality assessment: (1) *Random Selection* selects examples randomly. (2) *Response Length* employs the response length as the quality metric. (3) *Direct Scoring* (Chen et al., 2023) prompts ChatGPT to evaluate the accuracy of the response to the instruction directly. The associated prompt is displayed in Appendix E.3.

**Evol Quality:**   In a manner akin to EVOL COMPLEXITY, we introduce EVOL QUALITY to augment the discernment of quality measurement. For a given data sample $(I_k^{(0)}, R_k^{(0)})$, we prompt ChatGPT to elevate the quality of the response in an evolved way (Appendix E.4 for the prompt). This primarily involves enhancing helpfulness, augmenting relevance, enriching depth, fostering creativity, and supplying additional details. After $M$ iterations, for the same instruction $I_k^{(0)}$, we procure a set of responses spanning various qualities for $R_k$, denoted as $\{R_k^{(0)}, \cdots, R_k^{(M)}\}$. Similar to EVOL COMPLEXITY we set $M$ to 5.

As demonstrated in the middle part of Figure 1, we then instruct ChatGPT to rank and score these responses in terms of the response quality, thereby obtaining a quality score $q$ corresponding to each response (refer to the Appendix E.4 for the prompt template). Similar to *Evol Complexity*, this scoring approach is able to aid ChatGPT in discerning subtle differences between responses of varying qualities, and thus offering a more nuanced distinction of quality scores, as exemplified in Appendix B. We utilize scores derived from the seed dataset to fine-tune LLaMA-1 7B, enabling it to predict quality scores based on the provided instruction-response pair. The seed dataset is the same as the one for EVOL COMPLEXITY with 2K random samples from the Alpaca dataset.

**Results:**   Table 3 presents the experimental results of selecting the top 6K data from $X_{sota}$ and $X_{base}$ respectively. The proposed EVOL QUALITY approach consistently exhibits superior alignment performance. We notice that the pool with a higher quality variance, $X_{base}$, is more influenced by quality metrics, which is intuitive since many low-quality examples are present in such pools and will hurt the performance significantly. This result implies that quality is a necessary dimension to consider especially when working with data pools that have a substantial number of low-quality examples. We also observe that the response length positively correlates with the final alignment performance, yet its effect is not substantial for datasets with already high quality, such as $X_{sota}$

| Model | $X_{sota}$ | $X_{base}$ |
|---|---|---|
| Random Selection | 5.84 | 4.93 |
| Response Length | 5.94 | 5.65 |
| Direct Scoring (Pool=50K) | 5.61 | 4.44 |
| EVOL QUALITY (Pool=50K) | 5.85 | 5.29 |
| EVOL QUALITY | 6.19 | 5.67 |

Table 3: MT-bench of different quality measurements. All methods select 6K samples for training. "Pool=50K" denotes the data selection procedure is conducted in a 50K-sized subset due to the cost of ChatGPT to annotate the entire pool. We include the results of our method on the 50K data pool to make a fair comparison with the baselines.

| Model | $X_{sota}$ | $X_{base}$ |
|---|---|---|
| Random Selection | 5.82 | 4.34 |
| Instag Diversity | 6.10 | 4.46 |
| Repr Filter | 6.17 | 4.68 |

Table 4: MT-bench scores of different diversity measurements. All methods select 6K samples for instruction tuning.

## 2.5 FROM THE DIVERSITY PERSPECTIVE – AN EMBEDDING-BASED APPROACH

As a general principle, an advanced LLM should be adept at handling various requests from humans. Therefore, the data used for instruction tuning is desirable to maintain maximum diversity. However, real-world data often exhibits redundancy (Abbas et al., 2023). In this study, we explore the impact of data diversity on alignment by conducting a controlled experiment and then introduce a simple yet effective strategy $\pi_{diversity}$ to maintain diversity and conciseness in the selected subset of data.

**Setup:** In this work, we propose an iterative method to ensure the diversity of selected data. The iterative method picks sample $x_i$ from the pool $X$ one by one to selected dataset $S$ when $x_i$ contributes diversity to $S$. The process continues until the budget $m$ is reached or all $x_i$ in $X$ have been enumerated. To clarify, we formulate the benefit of diversity brought by a newly considered sample $x_i$ as an indicator function $\mathbb{1}[\mathcal{F}(x_i, S)]$ which is equal to 1 only if $\mathcal{F}(x_i, S)$ is True and 0 otherwise. $\mathcal{F}$, which we will define later, is the function to assess whether $x_i$ exhibits diversity in relation to the selected dataset $S$. Only when $\mathbb{1}[\mathcal{F}(x_i, S)]$ equals to 1, will $x_i$ be added to $S$. Additional setup details are in Appendix A.

**Baselines:** In addition to random selection, we further assess *Instag diversity* (Lu et al., 2023), which is designed iteratively to ensure diversity within the selected dataset. It utilizes the growth of the tag set for $S$ as a metric to define the function $\mathcal{F}$. Specifically, it formulates $\mathcal{F}_t = |T_S \bigcup T_{x_i}| > |T_S|$, where $T_S$ represents the set of all tags in $S$, and $T_{x_i}$ represents the tags associated with $x_i$.

**Repr Filter:** We examine our embedding-based approach, which we refer to as *Repr Filter*. Specifically, we take the distance between sample $x_i$ and its nearest neighbor in $S$ as the metric to define $\mathcal{F}$. We leverage the LLaMA-1 13B model to encode the sentence and compute the cosine distance $d$, then $\mathcal{F} := d < \tau$, where $\tau \in (0, 1)$ is a threshold hyperparameter. This means that we consider an example $x_i$ could increase the diversity of $S$ when the embedding distance between $x_i$ and its nearest neighbor is smaller than a threshold. During data selection, we first sort the data pool $X$ according to complexity and quality scores that we will detail in §3.1, then we examine each sample one by one and put $x_i$ to $S$ if $d < \tau$, where $S$ is initialized as empty. This process stops until the size of $S$ reaches the data budget $m$. We set threshold $\tau$ as 0.9 in all the relevant experiments across this paper, while we provide analysis results on $\tau$ and different sentence representations in Appendix C.1.

**Results:** Table 4 presents the results across different diversity strategies in $X_{sota}$ and $X_{base}$ respectively. Comparing Random Selection with the other two strategies ensuring diversity, the model trained with randomly selected data significantly underperforms others, which demonstrates the key role of diversity. And our approach is able to outperform Instag Diversity on both data pools.

## 3 DEITA– DATA EFFICIENT INSTRUCTION TUNING FOR ALIGNMENT

Based on our exploration in §2, we propose a simple approach to select data samples considering all the three dimensions, complexity, quality, and diversity. Utilizing the selected subset, we train our model DEITA , as an attempt towards extremely **D**ata-**E**fficient **I**nstruction **T**uning for **A**lignment. Below, we detail our data selection method and models.

### 3.1 METHOD

---

**Algorithm 1** Score-First, Diversity-Aware Data Selection

---

1: **Input:** The data pool $X$, data budget $m$
2: **Output:** The selected subset $S_{\pi_{\mathrm{DEITA}}}^{(m)}$
3: Initialize Empty Dataset $S_{\pi_{\mathrm{DEITA}}}^{(m)}$
4: Sorting $X$ with the combined complexity score and quality score $s = q * c$;
5: Getting the sorted Pool $X^*$;
6: **for** Each Sample $x \in X^*$ **do**
7:     // $d(x, S)$ denotes the distance between $x$ and its nearest neighbor in $S$
8:     **if** $d(x, S_{\pi_{\mathrm{DEITA}}}^{(m)}) < \tau$ **then**
9:         $S_{\pi_{\mathrm{DEITA}}}^{(m)} \leftarrow S_{\pi_{\mathrm{DEITA}}}^{(m)} \cup \{x\}$
10:     **else**
11:         Continue
12:     **end if**
13:     $X \leftarrow X \setminus \{x\}$
14:     **if** $|S_{\pi_{\mathrm{DEITA}}}^{(m)}|$ equals to $m$ **then**
15:         Break
16:     **end if**
17: **end for**

---

**Score-First, Diversity-Aware Data Selection:** While there are various ways of combining complexity, quality, and diversity measures, we aim to keep it as simple as possible to be practical. Intuitively, it is desirable to select samples with high complexity and quality scores but maintain the diversity of the set. To this end, we propose a score-first, diverse-aware data selection strategy, denoted as $\pi_{\mathrm{DEITA}}$. Our strategy incorporates a new evol score $s$ that combines complexity and quality by multiplying the complexity score $c$ with the quality score $q$ as $s := c * q$. For multi-turn dialogues, we calculate this score for each turn, summing them to obtain the final score for the entire conversation. Next, we sort all samples in $X$ using $s$, yielding the sorted pool $X^* = (x_1^*, x_2^*, ..., x_n^*)$, where $x_0^*$ represents the sample with the highest evol score. Beginning with $S_{\pi_{\mathrm{DEITA}}}^1 = (x_0^*)$, we iteratively select data from $X^*/S_{\pi_{\mathrm{DEITA}}}$ one by one, following the REPR FILTER strategy, and discard redundant samples for $S_{\pi_{\mathrm{DEITA}}}$. By integrating the evol score and the REPR FILTER, our approach guarantees complexity, quality, and diversity in the resulting dataset. Our data selection approach is illustrated in the right part of Figure 1 and summarized in Algorithm 1.

**Training DEITA:** We train DEITA using the selected dataset with $m$ samples. We denote the resulting models as $\mathrm{DEITA}_m$. In this paper, we train DEITA models based on LLaMA-1-13B, LLaMA-2-13B, and Mistral-7B respectively, and the training details are described in Appendix A.

## 3.2 EXPERIMENTAL SETUP

We train the DEITA models with data budget of 6K and 10K examples respectively. We select data samples from the data pool $X_{sota}$. We adopt MT-Bench, AlpacaEval, and the Open LLM Leaderboard as our benchmarks for automatic evaluation. The Open LLM Leaderboard consists of four classification tasks: ARC (Clark et al., 2018), HellaSwag (Zellers et al., 2019), MMLU (Hendrycks et al., 2021), and TruthfulQA (Lin et al., 2022). We also provide human evaluation results in Appendix D. We first take LLaMA-1-13B as the backbone and compare DEITA with other data selection approaches such as LIMA (Zhou et al., 2023), Alpagasus (Chen et al., 2023), and TAGLM (Lu et al., 2023). Then we compare DEITA models based on LLaMA-1-13B, LLaMA-2-13B, and Mistral-7B with other top-performing open-source models, such as Vicuna, WizardLM, Mistral-Instruct (Jiang et al., 2023), and Zephyr. In this paper, we mainly focus on models aligned using SFT without preference training since our contributions lie on data selection in the SFT stage, while we also run direct preference optimization (DPO, Rafailov et al. (2023)) on top of our best SFT model to report stronger performance for reference points. For DPO training, we randomly sample 10K comparison data pairs used in Zephyr that is originally obtained from the UltraFeedback dataset (Cui et al., 2023).

## 3.3 RESULTS

**Main Comparison:** We first compare DEITA with other data selection approaches in Table 5, where DEITA-LLaMA1$_{6K}$ clearly outperforms other approaches by a large margin. In Table 6, we train

| Model | Data Size | MT-Bench | AlpacaEval(%) |
|---|---|---|---|
| Random | 6K | 5.84 | 73.91 |
| Alpagasus (Pool=50K) | 6K | 5.61 | 71.21 |
| LIMA | 1K | 4.29 | 41.98 |
| TAGLM[†] | 6K | 6.09 | 72.80 |
| DEITA-LLaMA1-13B$_{6K}$ | 6K | **6.46** | **77.08** |

Table 5: Comparison of different data selection approaches, the backbone is LLaMA-1-13B. For Alpagasus, we could not use ChatGPT to score all the examples in the pool due to the cost, thus we score 50K random examples and select. † denotes the results obtained by using their released LLaMA-7B tagger model for a fair comparison.

| Model | Data Size / Alignment | MT-Bench | AlpacaEval(%) |
|---|---|---|---|
| **Proprietary Models** | | | |
| GPT-4 | – | 8.99 | 95.28 |
| Claude-v2 | – | 8.06 | 91.36 |
| gpt-3.5-turbo | – | 7.90 | 89.37 |
| **Open-sourced Models based on LLaMA-1-13B** | | | |
| Alpaca-13B | 52K / SFT | 4.53 | – |
| WizardLM-13B | 70K / SFT | 6.35 | 75.31 |
| Vicuna-13B-v1.3 | 125K / SFT | 6.39 | 82.11 |
| TAGLM-13B[†] | 6K / SFT | 6.09 | 72.80 |
| Random-Select | 10K / SFT | 6.03 | 71.52 |
| DEITA-LLaMA1-13B$_{6K}$ | 6K / SFT | 6.46 | 77.08 |
| DEITA-LLaMA1-13B$_{10K}$ | 10K / SFT | **6.60** | 78.01 |
| **Open-sourced Models based on LLaMA-2-13B** | | | |
| LLaMA2-13B-Chat | >100K / SFT + >1M / RLHF | 6.65 | 81.09 |
| Vicuna-13B-v1.5 | 125K / SFT | 6.57 | 78.80 |
| TÜLÜ 2 13B | 326K / SFT | 6.70 | 78.90 |
| TÜLÜ 2 + DPO 13B | 326K / SFT + 60K / DPO | 7.00 | 89.50 |
| Random-Select | 10K / SFT | 5.78 | 65.19 |
| DEITA-LLaMA2-13B$_{6K}$ | 6K / SFT | 6.65 | 80.75 |
| DEITA-LLaMA2-13B$_{10K}$ | 10K / SFT | **6.79** | **81.09** |
| **Open-sourced Models based on Mistral-7B** | | | |
| Mistral-7B-Instruct-v0.1 | – | 6.84 | 69.65 |
| Mistral-7B-Instruct-v0.2 | – | 7.60 | 93.65 |
| zephyr-beta-sft[*] | 200K / SFT | 5.32 | 75.12 |
| zephyr-beta | 200K / SFT + 60K / DPO | 7.34 | 90.60 |
| Random-Select | 10K / SFT | 5.89 | 56.90 |
| DEITA-Mistral-7B$_{6K}$ | 6K / SFT | 7.22 | 80.78 |
| DEITA-Mistral-7B$_{10K}$ | 10K / SFT | **7.32** | **81.67** |
| DEITA-Mistral-7B$_{6K}$ + DPO | 6K / SFT +10K / DPO | 7.55 | 90.06 |

Table 6: Results of different instruction-tuned models on MT-Bench and AlpacaEval. Best SFT-only numbers within the same base model are bolded, while the overall best numbers are underlined. † denotes the results obtained by using their released LLaMA-7B tagger model for a fair comparison. Zephyr-beta-sft[*] is the official checkpoint after the phase of supervised fine-tuning (SFT). We notice the performance of this checkpoint is lower than expected. We speculate the reason is that this checkpoint is not the best SFT checkpoint reported in their paper since the checkpoint is used for further DPO training.

LLaMA-1-13B, LLaMA-2-13B, and Mistral-7B with our selected 6K and 10K data respectively, and compare with other state-of-the-art SFT models as well as random data selection baselines (Random-Select). Across all three backbone models, the SFT aligned DEITA models outperform almost all other SFT-aligned models. DEITA models based on LLaMA-2 even outperforms LLaMA2-13B-Chat that undergoes RLHF training with carefully crafted human annotations. Notably, DEITA-Mistral-7B$_{10K}$ based on Mistral-7B achieves a 7.32 MT-Bench score, which is the state-of-the-art result among all open-source SFT models at 7B and 13B sizes. In the meanwhile, we note that the gains of DEITA on AlpacaEval are not apparently consistent with the gains on MT-Bench. To analyze such difference between MT-Bench and AlpacaEval, we plot the standard radar plot of the 8 substasks on MT-Bench, as shown in Figure 3. It clearly demonstrates that the DEITA Mistral models achieve high MT-Bench scores due to the enhanced performance on advanced abilities such as coding, math, and reasoning, which are not prominent in AlpacaEval. When equipped with DPO training, our DEITA-Mistral-7B$_{10K}$+DPO variant achieves 7.55 MT-Bench and 90.06% AlpacaEval scores, which is comparable to zephyr-beta that trains on 30x more data, and slightly lags behind the recent Mistral-7B-Instruct-v0.2 model whose alignment approach and data are not public.

| Model | Data Size / Alignment | ARC | HellaSwag | MMLU | TruthfulQA | Average |
|---|---|---|---|---|---|---|
| **Open-sourced Models based on LLaMA-1** | | | | | | |
| LIMA | 1K / SFT | 59.22 | 84.25 | 49.60 | 46.20 | 59.82 |
| WizardLM-13B | 70K / SFT | 57.25 | 80.88 | 52.92 | 50.55 | 58.96 |
| Vicuna-13B-v1.3 | 125K / SFT | 54.61 | 80.41 | 52.88 | 52.14 | 60.01 |
| Random-Select | 10K / SFT | 55.80 | 79.95 | 47.35 | 57.44 | 60.14 |
| DEITA-LLaMA1-13B$_{10K}$ | 10K / SFT | 59.47 | 82.01 | 60.60 | 55.03 | **64.27** |
| **Open-sourced Models based on LLaMA-2** | | | | | | |
| Vicuna-13B-v1.5 | 125K / SFT | 57.08 | 81.24 | 56.67 | 51.51 | 61.63 |
| Random-Select | 10K / SFT | 61.52 | 83.69 | 55.22 | 44.84 | 61.32 |
| DEITA-LLaMA2-13B$_{10K}$ | 10K / SFT | 58.87 | 82.08 | 55.33 | 54.57 | **62.71** |
| **Open-sourced Models based on Mistral-7B** | | | | | | |
| Mistral-7B-Instruct-v0.1 | – | 54.52 | 75.63 | 55.38 | 56.28 | 60.45 |
| zephyr-beta-sft | 200K / SFT | 57.68 | 81.98 | 61.04 | 43.00 | 60.93 |
| zephyr-beta | 200K / SFT + 60K / DPO | 62.03 | 84.52 | 61.44 | 57.44 | 66.36 |
| Random-Select | 10K / SFT | 55.38 | 79.16 | 58.73 | 53.59 | 61.72 |
| DEITA-Mistral-7B$_{6K}$ | 6K / SFT | 57.76 | 80.29 | 61.90 | 59.82 | 64.94 |
| DEITA-Mistral-7B$_{6K}$+DPO | 6K / SFT +10K / DPO | 66.21 | 85.42 | 60.66 | 67.14 | **69.86** |

Table 7: Results on the Open LLM Leaderboard. Data size by default represents the number of examples in SFT unless specified otherwise.

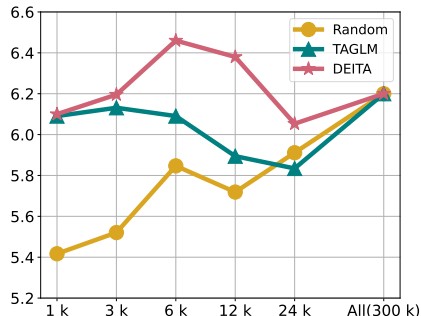

Figure 2: Data-scaling results on MT-Bench. The X-axis represents the # samples used.

Figure 3: Radar plot of detailed scores for DEITA models and major baselines on 8 subtasks on MT-Bench.

**Results on Open LLM Leaderboard:** We compare DEITA to other SOTA models and the random data selection baselines (Random-Select) on Open LLM Leaderboard, results are shown in Table 7. Our DEITA SFT models – while trained with only 6K or 10K data – achieve the best average results among SFT-aligned models across different backbones. Further DPO training greatly boosts the performance of DEITA-Mistra-7B by around 5 points on average and helps surpass the Zephyr.

**Data Scaling:** In order to investigate the data scaling effects of different data selection strategies, we experiment with subsets of $X_{sota}$ with various data budget $m$. Figure 2 illustrates that our DEITA models consistently deliver the best data selection performance across different data volumes. Remarkably, DEITA achieves comparable results to using all the 300K training samples with only 3K samples, a 100x data reduction. Interestingly, we find that under our data selection approach, as the quantity of selected data increases, the final performance initially shows an upward trend but eventually declines. This suggests that even for a relatively complex, diverse, and high-quality data pool like $X_{sota}$, the proportion of truly "good data for alignment" is limited. This phenomenon confirms that the performance of alignment does not necessarily improve even though we add more data and use more computing, implying the importance of data selection.

## 4 CONCLUSION

In this paper, we thoroughly investigate the question of what makes good data for alignment. Our research encompasses three controlled studies conducted across three dimensions: complexity, quality and diversity. Throughout these studies, we propose new methods for automatic data selection and train our models, DEITA, on the selected data. Experimental results demonstrate that DEITA is able to achieve superior or comparable performance to the state-of-the-art open-source models with 10x training samples. We release our selected data to effectively align models more efficiently.

## ACKNOWLEDGEMENT

This project is partially supported by the WeiXin Group in Tencent.

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

# A    Setup Details

**Diversity-based Data Selection:**    In the control experiment of §2.5, we meticulously manage the complexity and quality of the selected data to conduct a rigorous analysis of diversity's impact and facilitate a fair comparison between various methods. This is achieved by ensuring that the product of the mean complexity score $c$ and quality score $q$ of selected data $S$ closely align with the respective means of the complete data pool $X$, by limiting a deviation of a maximum value of 2 from the mean. Through this way, we first select a subset of samples from the data pool to construct a new pool, where the examples have similar $c * q$ scores. Then we explore different diversity selection approaches.

**Training Deita:**    We utilize four/eight NVIDIA Tesla A100 GPUs to train 7B/13B models. To facilitate parallel training, we employ DeepSpeed Zero-Stage 3 (Ren et al., 2021) and FlashAttention-2 (Dao, 2023). For the integration of multi-turn conversations, we use the Vicuna-style template. In all experiments of this paper, the training parameters are set with a maximum input length of 2048. In terms of Deita models based on LLaMA-1-13B, we set the batch size to 128, training epochs to 6, learning rates at 1e-5, and a warm ratio of 0.03. For Deita models based on LLaMA-2-13B, we set the batch size to 128, training epochs to 6, learning rate to 2e-5 and warm ratio to 0.1 follow Lu et al. (2023). Regarding Deita models built on Mistral-7B, we follow the hyperparameters from Tunstall et al. (2023) with a batch size 512, learning rate 2e-5, a warm ratio of 0.1 using cosine warmup scheduler for SFT, and batch size 32, learning rate 5e-7, a warm ratio of 0.1 using linear warmup scheduler for DPO. Due to the use of significantly less training data, to ensure adequate training, we increase the epochs for SFT training to 6 and the epochs for DPO to 9.

# B    Case Study

Table 8 and Table 9 display the instances of scoring complexity and quality using the Direct Scoring method and the Rank & Scoring method we proposed, respectively. An analysis of these instances indicates that Direct Scoring tends to assign similar and excessively high scores. In contrast, our method is capable of discerning the subtle variations in complexity or quality among different samples, thereby providing scores with distinct differentiation.

# C    Analysis

## C.1    Analysis of Repr Filter

We conduct experiments using various thresholds ($\tau$) ranging from 0.8 to 0.9 and different sentence encoding methods to assess their impact on the *Repr Filter*. We explore two methods for encoding sentences: one involves using the representation from the model which will be trained after data selection, while the other employs well-trained sentence embedding models such as E5-Large-V2 (Wang et al., 2022). We refer to the former method as the *Model-based* approach and the latter as the *Semantic-based* method. We utilize LLaMA-1 13B as the representation for the *Model-based* method and E5-Large-V2 to represent the *Semantic-based* method. For *Semantic-based*, we encode one sentence by averaging all embeddings of its tokens due to the limited context length for the model.

Figure 4 presents the results of our *Repr Filter* across different thresholds $\tau$ and different sentence representations. The findings demonstrate the robustness of the *Model-based* encoding method across different thresholds, maintaining a substantial margin over the baseline method. Conversely, the performance of the *Semantic-based* method exhibits a significant decline with adjustments to $\tau$. Due to the superior performance of E5-Large-V2 in MTEB benchmark (Muennighoff et al., 2023), we speculate (Bhatia et al., 2023) the representation from the *Model-based* method not only encodes semantic information within the sentences but also reflects certain characteristics of the data for subsequent instruction tuning, we prioritize its utilization.

# D    Human Evaluation

**Setup:**    Since it is difficult to conduct human evaluation on MT-Bench due to the challenging problems, we utilize the LIMA test dataset and randomly sample 100 of them as our evaluation

| Case A | | |
|---|---|---|
| **Instruciton** | **Direct Scroing** | **Ranking & Scoring** |
| Rate the given restaurant based on your experience. Restaurant: Pizzeria Garden | 4 | 2 |
| Using your experience as a guide, provide a comprehensive evaluation of Pizzeria Garden, taking into account factors such as the quality of the food, the level of service provided, the ambiance of the restaurant, and any other relevant aspects that may contribute to your overall impression. Please provide a detailed analysis that requires multiple steps of reasoning to arrive at a final rating. | 8 | 3 |
| Based on your extensive knowledge and expertise, please conduct a thorough and intricate assessment of Pizzeria Garden, encompassing various elements such as the excellence of the cuisine, the degree of attentiveness exhibited by the staff, the atmosphere and mood of the establishment, and any other pertinent factors that could influence your overall perception. Your evaluation should entail a meticulous analysis that necessitates multiple layers of reasoning to arrive at a conclusive rating. | 8 | 4 |
| In order to provide a thorough and nuanced evaluation of Pizzeria Garden, we require your extensive knowledge and specialized expertise. Your assessment should encompass a wide range of factors, including but not limited to the quality and inventiveness of the cuisine, the level of attentiveness and professionalism exhibited by the staff, the ambiance and overall atmosphere of the establishment, as well as any other pertinent elements that may influence your overall perception. Your analysis must be meticulous and multifaceted, necessitating a profound level of reasoning and critical thinking to arrive at a conclusive rating. | 8 | 5 |
| Case B | | |
| **Instruction** | **Direct Scroing** | **Ranking & Scoring** |
| Create a birthday wish for someone who loves animals. | 8 | 1 |
| Craft a heartfelt birthday message for an individual who has a deep affection for a specific type of animal, such as dogs, cats, birds, or reptiles. Incorporate their passion for this creature into your well wishes to make their special day even more memorable and meaningful. | 8 | 2 |
| Compose a sincere and thoughtful birthday greeting for an individual who holds a profound fondness for a particular species of fauna, be it canines, felines, avians, or cold-blooded creatures. Infuse their ardor for this living being into your felicitations to elevate their celebratory occasion to a higher level of significance and remembrance. | 7 | 4 |
| Compose a sincere and introspective birthday greeting for an individual who harbors a profound and enduring fondness for a particular species of fauna, be it domestic canines, felines, avians, or reptiles. Integrate their ardor for this animate being into your felicitations to amplify the significance and recollection of their festive occasion. Furthermore, contemplate incorporating individual reminiscences or minutiae about the selected genus to further individualize the message and showcase your comprehension and admiration of their passions. | 8 | 6 |

Table 8: Cases of complexity scores generated by different methods, including Direct Scoring and the Rank & Scoring method we proposed.

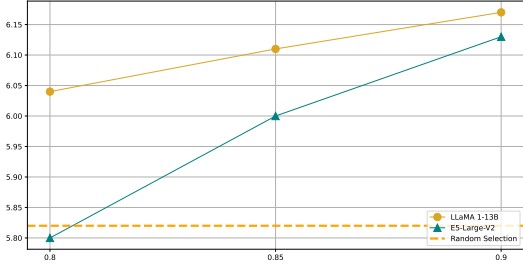

Figure 4: Results of setting different $\tau$ and leveraging different sentence representations in $X_{sota}$

prompts. Initially, we attempt to publish our evaluation on MTurk with a strict selection setting, as described in Li et al. (2023b). However, we encountered challenges as the answers from MTurkers

| Instruction A | | |
|---|---|---|
| Find the title of a book about a political movement in the 1940s. | | |
| **Response** | **Direct Scroing** | **Ranking & Scoring** |
| The title of a book about a political movement in the 1940s is "A Matter Of Resistance: The Struggle for Social Justice and Human Rights in India's 1940s". | 4.5 | 2 |
| One book that delves into the political movement of the 1940s is "A Matter Of Resistance: The Struggle for Social Justice and Human Rights in India's 1940s". This book provides a detailed account of the struggle for social justice and human rights during that time period. It sheds light on the challenges faced by the people and the efforts made to overcome them. The book is a valuable resource for anyone interested in understanding the political climate of the 1940s. | 4.5 | 3 |
| One book that delves into the political movement of the 1940s is "The Struggle for Social Justice and Human Rights in India\'s 1940s". This book offers a comprehensive account of the challenges faced by the people during that time period and the efforts made to overcome them. It provides valuable insights into the political climate of the 1940s, making it an excellent resource for anyone interested in gaining a deeper understanding of this era. The book is highly recommended for those who want to explore the political movement of the 1940s in greater depth. | 4.5 | 4 |
| The Struggle for Social Justice and Human Rights in India\'s 1940s" is a book that provides a detailed account of the political movement during that time period. It offers a comprehensive analysis of the challenges faced by the people and the efforts made to overcome them. The book provides valuable insights into the political climate of the 1940s, making it an excellent resource for anyone interested in gaining a deeper understanding of this era. It covers various aspects of the political movement, including the key players, their ideologies, and the impact of their actions. Overall, "The Struggle for Social Justice and Human Rights in India\'s 1940s" is a must-read for those who want to explore the political movement of the 1940s in greater depth. | 4.5 | 5 |
| Instruction B | | |
| Generate three new lines for an incomplete poem. | | |
| **Response** | **Direct Scoring** | **Ranking & Scoring** |
| That filled the morning air
Spreading joy and cheer
As far as one could see. | 4.5 | 2 |
| Here are three new lines to complete the poem:
The flowers bloomed in the meadow,
Their colors vibrant and bright,
A scene of beauty, a true delight. | 4.5 | 3 |
| Here are three new lines to complete the poem:
The birds chirped their melody
Welcoming the day with glee
Nature's beauty on full display. | 4.5 | 4 |
| The sun rose in the sky,
Its rays spreading warmth and light,
A new day dawning, full of promise and delight.
The birds joined in with their sweet melody,
A symphony of nature's harmony,
Welcoming the day with joy and glee.
And as the world awakens, we see,
Nature's beauty on full display,
A breathtaking sight, for all to see and say. | 4.5 | 5 |

Table 9: Cases of quality scores generated by different methods, including Direct Scoring and the Rank & Scoring method we proposed.

| DEITA-LLaMA1$_{6k}$ vs. | Win | Tie | Lose |
|---|---|---|---|
| Vicuna | 12% | 77% | 11% |
| Random Selection | 34% | 43% | 23% |

Table 10: Human evaluation results. Vicuna represents the Vicuna-13B-v1.3 model. Both DEITA and Random Selection use 6K training samples, while Vicuna is trained on 125K samples. The backbone model is LLaMA1-13B.

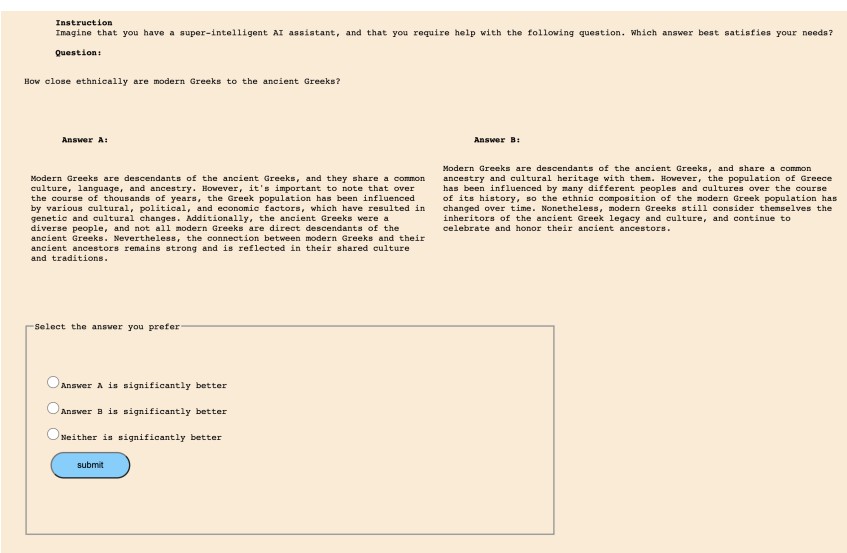

Figure 5: One example of our annotation prompts and interface.

exhibited inconsistency, even with the strict setting in place. Moreover, we observe difficulties in MTurkers following instructions well. Consequently, we enlist a group of 4 colleague researchers to serve as annotators for our evaluation. To ensure impartiality and conceal model bias, each annotator was assigned 50 samples without knowledge of the source of the two responses. Following the methodology outlined in Zhou et al. (2023), a single response was generated for each question. Throughout the evaluation process, annotators were presented with two responses from different models side by side and asked to indicate their preferred answer from three options: (1) Answer A is significantly better, (2) Answer B is significantly better, or (3) Neither is significantly better. The pairwise comparisons are conducted blindly, with the answer pairs randomly shuffled to conceal their sources. To ensure the reliability of our human evaluation, we performed the inter-annotator agreement test. In line with the methodology outlined by Zhou et al. (2023), we randomly selected 50 generation pairs from our evaluation and calculated the author-annotator agreement score using tie-discounted accuracy. Our findings yield an agreement score of 77%, which is reasonably high and aligns with the agreement levels reported in Zhou et al. (2023). The annotation interface and prompts are provided for reference in Figure 5.

**Results:** We compare DEITA-LLaMA1$_{6K}$ with the random selection baseline and Vicuna-13B-v1.3. In Table 10, we observe that human preferences align closely with GPT-4 scoring evaluations on MT-Bench. Our data selection strategy demonstrates a significant advantage over random selection in terms of human evaluation. DEITA-LLaMA1$_{6K}$ performs on par with Vicuna-13B-v1.3 in terms of human evaluation with most of the responses being considered tie by the annotators. However, we note that DEITA-LLaMA1$_{6K}$ is trained with 20x less data than Vicuna-13B-v1.3 which utilizes 125K training samples.

| | Prompt Template |
|---|---|
| **Direct Scoring** | We would like you to evaluate and rate the difficulty and complexity of the following question. You should give an overall score on a scale of 1 to 10, where a higher score indicates higher difficulty and complexity. You must just give a score without any other reasons.
Question: <Instruction>
Score: |
| **Instruction Node** | You need to rewrite the following "instruction" to a TREE through Semantic Parsin in the natural language processing field and only count the total node number of the TREE. You must just give node number without any other reasons.
Instruction: <Instruction>
Node number: |

Table 11: The corresponding Prompt Templates for the Complexity baseline, including Direct Scoring and Instruction Node.

| | Prompt Templates |
|---|---|
| **adding constraints** | I want you act as a Prompt Rewriter.
Your objective is to rewrite a given prompt into a more complex version to make those famous AI systems (e.g., ChatGPT and GPT4) a bit harder to handle.
But the rewritten prompt must be reasonable and must be understood and responded by humans.
Your rewriting cannot omit the non-text parts such as the table and code in #Given Prompt#:. Also, please do not omit the input in #Given Prompt#.
You SHOULD complicate the given prompt using the following method:
Please add one more constraints/requirements into #Given Prompt#
You should try your best not to make the #Rewritten Prompt# become verbose, #Rewritten Prompt# can only add 10 to 20 words into #Given Prompt#.
'#Given Prompt#', '#Rewritten Prompt#', 'given prompt' and 'rewritten prompt' are not allowed to appear in #Rewritten Prompt#
#Given Prompt#:
<Here is instruction>
#Rewritten Prompt#: |
| **deepening** | I want you act as a Prompt Rewriter.
Your objective is to rewrite a given prompt into a more complex version to make those famous AI systems (e.g., ChatGPT and GPT4) a bit harder to handle.
But the rewritten prompt must be reasonable and must be understood and responded by humans.
Your rewriting cannot omit the non-text parts such as the table and code in #Given Prompt#:. Also, please do not omit the input in #Given Prompt#.
You SHOULD complicate the given prompt using the following method:
If #Given Prompt# contains inquiries about certain issues, the depth and breadth of the inquiry can be increased. or
You should try your best not to make the #Rewritten Prompt# become verbose, #Rewritten Prompt# can only add 10 to 20 words into #Given Prompt#.
'#Given Prompt#', '#Rewritten Prompt#', 'given prompt' and 'rewritten prompt' are not allowed to appear in #Rewritten Prompt#
#Given Prompt#:
<Here is instruction>
#Rewritten Prompt#: |

Table 12: Prompt used to increase the complexity of the instruction, including adding constraints and deepening.

# E  PROMPT EXAMPLES

## E.1  COMPLEXITY BASELINE

Table 11 displays the prompts used by baselines such as Direct Scoring and Instruction Node as complexity metrics.

## E.2  EVOL COMPLEXITY

| | Prompt Templates |
|---|---|
| **concretizing** | I want you act as a Prompt Rewriter. 
 Your objective is to rewrite a given prompt into a more complex version to 
 make those famous AI systems (e.g., ChatGPT and GPT4) a bit harder to handle. 
 But the rewritten prompt must be reasonable and must be understood and 
 responded by humans. 
 Your rewriting cannot omit the non-text parts such as the table and code in 
 #Given Prompt#:. Also, please do not omit the input in #Given Prompt#. 
 You SHOULD complicate the given prompt using the following method: 
 Please replace general concepts with more specific concepts. or 
 You should try your best not to make the #Rewritten Prompt# become verbose, 
 #Rewritten Prompt# can only 
 add 10 to 20 words into #Given Prompt#. 
 '#Given Prompt#', '#Rewritten Prompt#', 'given prompt' and 'rewritten prompt' 
 are not allowed to appear in #Rewritten Prompt# 
 #Given Prompt#: 
 <Here is instruction> 
 #Rewritten Prompt#: |
| **increasing reasoning steps** | I want you act as a Prompt Rewriter. 
 Your objective is to rewrite a given prompt into a more complex version to 
 make those famous AI systems (e.g., ChatGPT and GPT4) a bit harder to handle. 
 But the rewritten prompt must be reasonable and must be understood and 
 responded by humans. 
 Your rewriting cannot omit the non-text parts such as the table and code in 
 #Given Prompt#:. Also, please do not omit the input in #Given Prompt#. 
 You SHOULD complicate the given prompt using the following method: 
 If #Given Prompt# can be solved with just a few simple thinking processes, 
 you can rewrite it to explicitly request multiple-step reasoning. 
 You should try your best not to make the #Rewritten Prompt# become verbose, 
 #Rewritten Prompt# can only 
 add 10 to 20 words into #Given Prompt#. 
 '#Given Prompt#', '#Rewritten Prompt#', 'given prompt' and 'rewritten prompt' 
 are not allowed to appear in #Rewritten Prompt# 
 #Given Prompt#: 
 <Here is instruction> 
 #Rewritten Prompt#: |

Table 13: Prompt used to increase the complexity of the instruction, include concretizing and increasing reasoning steps.

| | Prompt Templates |
|---|---|
| **Rank & Score** | Ranking the following questions according to the difficulty and complexity. Score 1-5. 
 You can give a score of 6 if the question is too complex for you to answer it. You should 
 respond with the format:\n [1] Score: 1\n [2] Score: 2\n 
 [1] <Instruction 1> 
 [2] <Instruction 2> 
 [3] <Instruction 3> 
 [4] <Instruction 4> 
 [5] <Instruction 5> |

Table 14: Prompt used to rank and score instructions of different complexities.

**Prompt for Enhancing Complexity**  Table 12 and Tabel 13 display the prompts used to enhance the complexity such as adding constraints, deepening, concretizing and increasing reasoning steps..

**Prompt for Ranking and Scoring**  Table 14 shows the prompts used to rank and score instructions of different complexities.

### E.3  QUALITY BASELINE

Table 15 displays the prompts used by baselines such as Direct Scoring as quality metrics.

| | Prompt Templates |
|---|---|
| **Direct Scoring** | We would like to request your feedback on the performance of AI assistant in response to the given question displayed following.
##Tips:Please rate according to the accuracy of the response to the instruction and the input. Each assistant receives a score on a scale of 0 to 5, where a higher score indicates higher level of the accuracy. You must just give a score without any other reasons.
##Question:
<Instruction>
##Response:
<Response>
##Score: |

Table 15: The corresponding Prompt Templates for the Quality baseline, including Direct Scoring.

### E.4 EVOL QUALITY

**Prompt for Enhancing Quality**   Table 16 and Table 17 display the prompts to enhance the quality such as enhancing helpfulness, augmenting relevance, enriching depth, fostering creativity and supplying additional details.

**Prompt for Ranking and Scoring**   Table 18 shows the prompts used to rank and score instructions of different qualities.

| | Prompt Templates |
|---|---|
| **enhancing helpfulness** | I want you to act as a Response Rewriter
Your goal is to enhance the quality of the response given by an AI assistant
to the #Given Prompt# through rewriting.
But the rewritten response must be reasonable and must be understood by humans.
Your rewriting cannot omit the non-text parts such as the table and code in
#Given Prompt# and #Given Response#. Also, please do not omit the input
in #Given Prompt#.
You Should enhance the quality of the response using the following method:
Please make the Response more helpful to the user.
You should try your best not to make the #Rewritten Response# become verbose,
#Rewritten Response# can only add 10 to 20 words into #Given Response#.
'#Given Response#', '#Rewritten Response#', 'given response' and 'rewritten response'
are not allowed to appear in #Rewritten Response#
#Given Prompt#:
Give three tips for staying healthy.
#Given Response#:
<Response>
#Rewritten Response#: |
| **augmenting relevance** | I want you to act as a Response Rewriter
Your goal is to enhance the quality of the response given by an AI assistant
to the #Given Prompt# through rewriting.
But the rewritten response must be reasonable and must be understood by humans.
Your rewriting cannot omit the non-text parts such as the table and code in
#Given Prompt# and #Given Response#. Also, please do not omit the input
in #Given Prompt#.
You Should enhance the quality of the response using the following method:
Please make the Response more relevant to #Given Prompt#.
You should try your best not to make the #Rewritten Response# become verbose,
#Rewritten Response# can only add 10 to 20 words into #Given Response#.
'#Given Response#', '#Rewritten Response#', 'given response' and 'rewritten response'
are not allowed to appear in #Rewritten Response#
#Given Prompt#:
Give three tips for staying healthy.
#Given Response#:
<Response>
#Rewritten Response#: |
| **enriching depth** | I want you to act as a Response Rewriter
Your goal is to enhance the quality of the response given by an AI assistant
to the #Given Prompt# through rewriting.
But the rewritten response must be reasonable and must be understood by humans.
Your rewriting cannot omit the non-text parts such as the table and code in
#Given Prompt# and #Given Response#. Also, please do not omit the input
in #Given Prompt#.
You Should enhance the quality of the response using the following method:
Please make the Response more in-depth
You should try your best not to make the #Rewritten Response# become verbose,
#Rewritten Response# can only add 10 to 20 words into #Given Response#.
'#Given Response#', '#Rewritten Response#', 'given response' and 'rewritten response'
are not allowed to appear in #Rewritten Response#
#Given Prompt#:
Give three tips for staying healthy.
#Given Response#:
<Response>
#Rewritten Response#: |

Table 16: Prompt used to increase the quality of the response, include enhancing helpfulness, augmenting relevance and enriching depth.

| | Prompt Templates |
|---|---|
| **fostering creativity** | I want you to act as a Response Rewriter
Your goal is to enhance the quality of the response given by an AI assistant
to the #Given Prompt# through rewriting.
But the rewritten response must be reasonable and must be understood by humans.
Your rewriting cannot omit the non-text parts such as the table and code in
#Given Prompt# and #Given Response#. Also, please do not omit the input
in #Given Prompt#.
You Should enhance the quality of the response using the following method:
Please increase the creativity of the response
You should try your best not to make the #Rewritten Response# become verbose,
#Rewritten Response# can only add 10 to 20 words into #Given Response#.
'#Given Response#', '#Rewritten Response#', 'given response' and 'rewritten response'
are not allowed to appear in #Rewritten Response#
#Given Prompt#:
Give three tips for staying healthy.
#Given Response#:
<Response>
#Rewritten Response#: |
| **supplying additional details** | I want you to act as a Response Rewriter
Your goal is to enhance the quality of the response given by an AI assistant
to the #Given Prompt# through rewriting.
But the rewritten response must be reasonable and must be understood by humans.
Your rewriting cannot omit the non-text parts such as the table and code in
#Given Prompt# and #Given Response#. Also, please do not omit the input
in #Given Prompt#.
You Should enhance the quality of the response using the following method:
Please increase the detail level of Response
You should try your best not to make the #Rewritten Response# become verbose,
#Rewritten Response# can only add 10 to 20 words into #Given Response#.
'#Given Response#', '#Rewritten Response#', 'given response' and 'rewritten response'
are not allowed to appear in #Rewritten Response#
#Given Prompt#:
Give three tips for staying healthy.
#Given Response#:
<Response>
#Rewritten Response#: |

Table 17: Prompt used to increase the quality of the response, include fostering creativity and supplying additional details.

| | Prompt Templates |
|---|---|
| **Rank & Score** | Rank the following responses provided by different AI assistants to the user's question
according to the quality of their response. Score each response from 1 to 5, with 6
reserved for responses that are already very well written and cannot be improved further.
Your evaluation should consider factors such as helpfulness, relevance, accuracy, depth,
creativity, and level of detail of the response.
Use the following format:
[Response 1] Score:
[Response 2] Score:
#Question#: <Instruction>
#Response List#:
[Response 1] <Response 1>
[Response 2] <Response 2>
[Response 3] <Response 3>
[Response 4] <Response 4>
[Response 5] <Response 5> |

Table 18: Prompt used to rank and score responses of different qualities.

