# OpenReview forum: "What Makes Good Data for Alignment? A Comprehensive Study of Automatic Data Selection in Instruction Tuning"
_ICLR.cc/2024/Conference — ICLR 2024 poster_

### Official Review · Reviewer_RtKh · 2023-10-28

**Soundness:** 2 fair
**Presentation:** 3 good
**Contribution:** 2 fair
**Rating:** 5
**Confidence:** 2

**Summary:**

This paper investigates the trendy problem of selecting high-quality instructions for fine-tuning pre-trained language models (PLMs). The paper aims to provide an automated pipeline for this selection problem. Following recent works, sample quality is evaluated on three dimensions: complexity, quality, and diversity where this evaluation is conducted by other LMs such as GPT-4. The paper validates the proposed approach with LLaMA-1 and shows that when selecting samples from a larger pool of lower-quality data, the proposed method, DEITA (Data-Efficient Instruction Tuning for Alignment), is able to match the performance of current open-source alignment models with a small fraction of fine-tuning data.

**Strengths:**

The paper is, in general, an easy read. Its ideas are presented straightforwardly along with nice tables and illustrations.

The problem it investigates is trendy. Quality for each part of the work is generally fine–it is structurally complete, self-contained, and has a reasonable narrative.

**Weaknesses:**

This paper is "ok" but not particularly enticing. The topic is trendy but the approach is not technically challenging or particularly innovative.

I have recently read a number of papers on this topic of instruction mining with LLaMA/Alpaca–to name a few

- a. ALPAGASUS : TRAINING A BETTER ALPACA WITH FEWER DATA
- b. Instruction Mining: High-Quality Instruction Data Selection for Large Language Models
- c. InstructionGPT-4: A 200-Instruction Paradigm for Fine-Tuning MiniGPT-4

The technical body of these papers are uncomfortably similar. Combining a number of existing metrics (often in trivial ways) such as quality, diversity, etc. as a new evaluation metric and conducting the evaluations largely with the help of GPTs. And the end goals are also the same–to achieve comparable or better performance with fewer samples. I'm not sure how much "research gap" remains there and how individual works may continue to contribute to that–at least, this concern is still not resolved by this paper.

In terms of quality, the construction of this paper is simple. It does not have many ablation studies or insightful discussions of design choices/novel findings. No additional results or further discussions are provided in the Appendix.

It is an "okay" paper without much particular. Given its position in this apparently overly populated track, I would not be very interested in seeing it at the conference and vote against publishing it.

- Reproducibility: Code, data, model checkpoints, or data selection tools are not provided during the reviewing phase.

- Format: Appendix is not cut from the main paper. The PDF provided for the main paper is this 20-page document.

**Questions:**

If the authors wish to further develop this work toward publication at a future venue, I suggest the authors to

1. spend significant effort in discussing the current research landscape and identify a valid (important and essential) research gap that this work will make a substantial and unique contribution to. This is my main concern for this line of work.

2. improve the quality of the paper. If its extent of innovation is capped by the many other papers out there,  this work needs to have a high quality to be a valid contribution. This includes conducting more experiments and ablation studies and discussing design choices and novel findings.

3. improve the technical approach. The current methods documented in this paper do not seem particularly novel compared to existing works and its technical contributions seem capped by the heavy reliance on GPTs. If the authors could design original technical approaches (new metrics, new evaluation methods, novel ways for combining different metrics), the paper could be made much more attractive.

- Appendix should not be submitted under the main paper.

---

> ### Author Response · Authors · 2023-11-20
> **Response 1/2**
>
> Thank you for the time and comments, we have enriched our evaluation with additional strong results on more benchmarks with different backbone models, please refer to the General Response for details. We address your separate concerns below.
>
> ### Q1: How much “research gap” remains in instruction mining or SFT data selection? What is the research gap that this work contributes to?
>
> We respectfully disagree with the reviewer on these comments about this research line in general, as well as our work’s contribution and significance. While we have no doubt that the reviewer understands these “simple” methods from previous works and our study, we believe there may be an oversight from the reviewer regarding the overall progress and current state of this research direction.
>
> To provide context, we would like to briefly summarize recent advancements in SFT data selection, which will illustrate the "research gap" and the broader picture. Initial studies like Self-instruct [1] and Alpaca [2] showed that medium-sized datasets (ranging from 50k to 100k samples) could endow pretrained models with effective instruction-following capabilities, though Alpaca was later identified as a weak baseline upon more rigorous evaluations. LIMA [3] highlighted the significance of data selection in SFT by using just 1,000 carefully chosen or human-crafted samples to surpass Alpaca's performance. However, these samples were not selected automatically.
>
> Alpagasus [4] and Instruction Mining [5], as the reviewer mentioned, began exploring automatic data selection in SFT. Yet, their comparisons were limited to random baselines or the already weak model Alpaca at the time. They did not conduct standard evaluations on challenging benchmarks like MT-bench/AlpacaEval nor did they compete with strong models such as Vicuna and WizardLM. This indicates a clear gap: **no study in automatic data selection has demonstrated practical effectiveness through evaluations on challenging benchmarks and against SOTA open-source SFT models -- we do not know whether [4] and [5] really work or not from their weak evaluations**. This gap is significant as it leaves unanswered questions about the potential and practicality of automatic data selection and whether it can lead to SOTA open-source SFT models.
>
> We note that [4] and [5] have updated their papers with more evaluations and baselines *after* the ICLR submission deadline, which we hope the reviewer realizes. However, even in their latest versions, [4] does not compare with any other state-of-the-art open-source SFT model. Our approach has significantly outperformed it (as shown in Table 5 of our paper). Meanwhile, [5] does not report standard MT-bench and AlpacaEval scores.
>
>
> ### Q2: Our contribution and significance over previous/recent works on instruction mining.
>
> Following the response above, our research stands out as the *first* to propose an automatic data selection approach that consistently demonstrates strong performance across multiple challenging benchmarks, including MT-bench, AlpacaEval, and the Open LLM Leaderboard. Our approach outperforms other data selection methods by a large margin (Table 5), yielding results that are comparable to, or even surpass, those of SOTA open-source SFT models such as Vicuna, WizardLM, and Mistral-Instruct. These empirical results – being the first to achieve SOTA open-source performance with 10x less data selected automatically –  is particularly noteworthy because it suggests that our approach, the datasets selected, and the data selection tools we have developed can be practically and beneficially applied by a broad spectrum of researchers and practitioners.
>
> In terms of technical innovation, our approach diverges from previous methodologies. We utilize an evolving strategy to elicit more faithful complexity and quality scores from ChatGPT, as opposed to a direct scoring method, which our results (shown in Tables 2 and 3) indicate is less effective. While we recognize that the combination of metrics and reliance on GPT models are not novel, we have no reason to complicate things when we find such simple or trivial combinations of our proposed metrics are highly effective. Methodologically, exploring alternatives to GPT reliance is an intriguing future direction, yet it was not deemed essential for the current scope of this paper, particularly given that our data selection tool does not require ChatGPT at the inference stage.
>
>
> Furthermore, we would like to remind the reviewer that both Alpagasus and InstructionMining were first arxived within 3 months prior to the ICLR submission deadline. According to the ICLR official reviewer guide (https://iclr.cc/Conferences/2024/ReviewerGuide#Reviewing%20instructions), such works are regarded as concurrent. While we still believe our work represents significant advancements beyond these studies as described above, we urge the reviewer to consider this timing factor as well when evaluating our research.

---

> ### Author Response · Authors · 2023-11-20
> **Response 2/2**
>
> ### Q3: Reproducibility: Code, data, model checkpoints, or data selection tools are not provided during the reviewing phase.
>
> We have promised to release these resources after the review period in the footnote of the first page, we surely will do when we deanonymize this paper – it is not common practice to release model checkpoints in an anonymous way during the review period as far as we know.
>
>
> ### Q4: Appendix is not cut from the main paper. Appendix should not be submitted under the main paper.
>
> > Authors may use as many pages of appendices (*after the bibliography*) as they wish
>
> quoted from ICLR 2024 Call for Papers (https://iclr.cc/Conferences/2024/CallForPapers).
>
> > Q: Should the appendices be added as a separate PDF or in the same PDF as the main paper?
> Either is allowed: you can include the appendices at the end of the main pdf after the references, or you can include it as a separate file for the supplementary materials.
>
> quoted from ICLR 2024 Author Guide (https://iclr.cc/Conferences/2024/AuthorGuide)
>
>
> [1] Wang et al. Self-Instruct: Aligning Language Models with Self-Generated Instructions. https://aclanthology.org/2023.acl-long.754.pdf ACL 2023.
> [2] Taori et al. Stanford Alpaca: An Instruction-following LLaMA model. https://crfm.stanford.edu/2023/03/13/alpaca.html 2023.
> [3] Zhou et al. LIMA: Less Is More for Alignment. https://arxiv.org/abs/2305.11206. 2023.
> [4] Chen et al. AlpaGasus: Training A Better Alpaca with Fewer Data. https://arxiv.org/abs/2307.08701. 2023.
> [5] Cao et al. Instruction Mining: When Data Mining Meets Large Language Model Finetuning. https://arxiv.org/pdf/2307.06290.pdf 2023.

---

> ### Author Response · Authors · 2023-11-22
> **A Kind Reminder for Reading the Response**
>
> Dear Reviewer RtKh,
>
> We have revised the paper and included additional strong results on more benchmarks. In the response above, we have also tried to clarify progresses on this research direction and our contributions over previous (concurrent) works. Since the rebuttal period is closing very soon, could you please check our response to see whether it mitigates your concerns? We would greatly appreciate that!
>
> Thank you,
> The authors

---

> > ### Comment · Reviewer_RtKh · 2023-11-23
> >
> > Thanks to the authors for the careful response. The explanations and additional results definitely helped to better contextualize this work. I can understand the challenge of working on a trending topic with new results coming out all the time. From the author's explanation, the performance achieved by this work is at the frontier.
> >
> > The part of my review remains that I do not see a clear technical contribution offered by this work, at least not a principled framework. I am not very familiar with the state-of-the-art performance for instruction tuning so I will have to leave the evaluation of this part to other reviewers and the AC.
> >
> > Thus, I am not passionately against the publication of the work or passionately supporting it. I would raise my score to 5 while decreasing my confidence to 2.

---

### Official Review · Reviewer_zdke · 2023-11-01

**Soundness:** 3 good
**Presentation:** 4 excellent
**Contribution:** 2 fair
**Rating:** 6
**Confidence:** 3

**Summary:**

This work shows that complexity, quality, and diversity are all important for data selection in instruction tuning. Based on the studies, it proposes a score-first, diversity-aware approach called DEITA to select the “good” data samples. With their proposed learning-based enhanced data measures, DEITA shows better or on par performance with state-of-the-art LLMs with only 6K training samples.

**Strengths:**

- The paper provides in-depth controlled studies to show the proposed scoring measurements are better than baselines in terms of complexity and quality.
- With fine-tuned LLaMA-7B models on the 20K Alpaca dataset with the scores from GPT, the measurements can score unseen instructions at a cheap cost.
- The proposed diversity-aware selection method is efficient and easy to implement.
- Clear presentation and easy-to-follow writing.

**Weaknesses:**

Limited evaluation
- No evaluation result on benchmarks like MMLU and Big-Bench-Hard, which allows verification of commonsense knowledge and reasoning with ground-truth answers. GPT4-based evaluation often includes errors or bias, so not enough.
- Marginal performance improvement. In Table 3, DEITA is worse than Vicuna-13B on AlpacaEval dataset. Also, In Table 4 with human evaluation, Vicuna performs almost similar to DEITA.
- Limited baselines; recent instruction selection works including LIMA and Alpagasus are missing in the evaluation.

**Questions:**

Why the complexity measurement is obtained from “instruction”, and the quality measurement is obtained from “response”? Is there any intuition of this?

Typo: “large margine” should be“large margin” in the last sentence of the introduction.

---

> ### Author Response · Authors · 2023-11-20
> **Response**
>
> Thanks for your helpful comments! Below we address your concerns.
>
> ### Q1: Limited evaluation
>
> Thanks for the comments. We have added more evaluation results on the Open LLM Leaderboard benchmark [1] that consists of four tasks, ARC, HellaSwag, MMLU, and TruthfulQA.  Additionally, we have enriched our evaluations by training more pre-trained LLMs on our selected data, such as  LLaMA-2-13B and Mistral-7B besides LLaMA-1-13B in the original submission. We have summarized these results in the table of the general response, while the detailed results are in Table 3,4,5 of the paper. On the Open LLM Leaderboard, our approach outperforms the random baseline and other strong models across three different backbone models. However, we would like to note that the performance on benchmarks such as MMLU is largely determined by the underlying base model, thus the gains after alignment tuning is typically not large, unless the training data is specifically crafted for the benchmark. This is why recent works on alignment put more emphasis on MT-bench and AlpacaEval [2].
>
> Notably, on MT-bench and AlpacaEval, our added results show that our method with the LLaMA-2-13B model surpasses Vicuna-13B-1.5 that trains on 10x more data, and LLaMA-2-13B Chat that has undergone extensive RLHF training. With Mistral-7B, our approach attains an MT-bench score of 7.29 with only 10k training data, marking a state-of-the-art performance on MT-bench *for all open-source SFT models at 7B and 13B sizes*. We also observe that, while the gains of the Mistral DEITA model on AlpacaEval are slightly lower than those on MT-bench, a deeper analysis (as depicted in Figure 2) indicates that the Mistral DEITA model excels particularly in mathematical and complex reasoning tasks, areas not primarily covered by AlpacaEval.
>
> ### Q2: Marginal performance improvement. DEITA is worse than Vicuna-13B on AlpacaEval dataset. With human evaluation, Vicuna performs almost similar to DEITA.
>
> We emphasize that the purpose of this work is not to set up a new record on the performance, but to achieve top results with much less data, for example, the training of DEITA is far more efficient than Vicuna by using 10x less training data. Furthermore, we note that Vicuna is a SOTA SFT model, and DEITA outperforms it on most of the cases across three different benchmarks and various backbone models (see the table in General Response), except for just one setting on AlpacaEval –  **DEITA is the first automatic data selection approach that achieves comparable performance to SOTA models such as Vicuna and WizardLM with 10x less data**. Therefore, we view comparable performance to Vicuna as the strength of our approach, rather than weaknesses.
>
>
>
> ### Q3: Missing comparison with instruction selection works
> Thanks for the suggestion! In the original submission,  we have shown that “direct scoring” from Alpagasus [3] does not work well in Table 2 and 3, and using ChatGPT to score all the samples in the data pool following their paper is too expensive, thus we did not explicitly include them in the main tables. Following the reviewer’s advice, we have added the Alpagasus and LIMA baselines, the comparison of different instruction selection approaches is shown below (based on LLaMA-1-13B):
>
> | **Model**                  | **Data Size** | **MT-Bench** | **AlpacaEval(%)** |
> |----------------------------|---------------|--------------|-------------------|
> | Random                     | 6K            | 5.84         | 73.91             |
> | Alpagasus      | 6K            | 5.61         | 71.21                |
> | LIMA                       | 1K            | 4.29         | 41.98             |
> | $\text{TagLM}$  | 6K            | 6.09         | 72.80             |
> | $\text{DEITA}_{6K}$              | 6K            | **6.46**     | **77.08**         |
>
>
>
> We see that DEITA outperforms other instruction selection baselines significantly. We have included the table above as Table 5 in the paper.
>
>
> ### Q4: Why the complexity measurement is obtained from “instruction”, and the quality measurement is obtained from “response”?
>
> We measure complexity from instruction only since the evolution procedure of complexity directly follows WizardLM, which takes only the instruction as input and evolves it step by step to increase the instruction complexity.  For quality, our intuition is to measure how effectively a response addresses the query, thus both the instruction and the response are inputs to the scorer. We recognize that in Figure 1, the instruction input was inadvertently omitted in the depiction of the quality scorer. This was our oversight, we have slightly revised Figure 1 to be more clear on this.
>
>
>
> [1] Beeching et al. Open LLM Leaderboard. https://huggingface.co/spaces/HuggingFaceH4/open_llm_leaderboard. 2023.
> [2] Tunstall et al. Zephyr: Direct Distillation of LM Alignment. https://arxiv.org/abs/2310.16944. 2023

---

> > ### Comment · Reviewer_zdke · 2023-11-21
> > **Rebuttal response**
> >
> > Thanks to the comprehensive rebuttal. The authors addressed most of my concerns with more results on Open LLM Leaderboard. I will increase my score to 6.

---

### Official Review · Reviewer_NSy4 · 2023-11-01

**Soundness:** 2 fair
**Presentation:** 3 good
**Contribution:** 3 good
**Rating:** 8
**Confidence:** 4

**Summary:**

This paper presents a data selection algorithm for instruction tuning that selects data points where 1) the queries are complex (e.g.: in terms of the constraints in the request); 2) the responses are high quality (e.g.: helpfulness, creativity); and 3) the data points are diverse. For quantifying complexity and quality, a dataset is collected for each of the two aspects by prompting ChatGPT in the following way: a seed set of samples are taken from the original dataset, ChatGPT is prompted to iteratively improve the complexity or quality of each of those samples along relevant dimensions, ChatGPT is then asked to score these samples for complexity or quality, a separate model (Llama-7B) is trained to predict these scores and then used to score the entire instruction tuning dataset. For diversity, the selected pool (initialized to be empty) is iteratively grown by adding points only if they are beyond a certain distance to their nearest neighbors already in the pool.

The data selection procedure sorts the existing dataset by the product of quality and complexity scores, and uses the diversity based selection procedure to select the points from the sorted set to a prespecified size.

The algorithm is evaluated by comparing Llama-13B models trained using the data selection procedure against existing instruction tuned models (trained on other datasets), and random selection baselines (trained on the same datasets of the same sizes with points selected randomly). Comparisons are made in terms of AlpacaEval and MT-Bench, where GPT-4 evaluates the responses and human evaluation on a sample of 100 requests from LIMA's test set. The algorithm outperforms random selection, and also other instruction tuned models based on Llama-1 trained on more data.

**Strengths:**

- The procedure used to quantify complexity and quality is innovative and can be used for other hard to quantify aspects of data quality in future work.
- The results from the experiments clearly show that the proposed method is indeed selecting important data points (i.e., better than random selection) at least for improving performance on AlpacaEval and MT-Bench.

**Weaknesses:**

The evaluation in this paper is limited and leaves some important questions unanswered:

- The main evaluation is done in terms of AlpacaEval and MT-Bench alone. Since these are relatively small evaluation sets and it has been shown that GPT-4 evaluation can be biased (Wang et al., 2023; https://arxiv.org/abs/2306.04751), one wonders if the data selection does better than random only because it is aligned with those biases. Including further evaluation, possibly on targeted benchmarks covering abilities like reading comprehension, complex reasoning etc. can be helpful.
- Related to the above point, since human evaluation was done on only 100 instances, it would be helpful to quantify the reliability of this assessment, e.g.: using inter-annotator agreement scores and significance, and perform a larger scale evaluation if needed.

The procedure used for quantifying complexity and quality can be validated and possibly improved further

- Section 2.3 states that ChatGPT is shown multiple samples evolved from the same seed example are shown to ChatGPT at a time for scoring their complexity (and this is possibly true for quality as well). Are these scores comparable across evalved sets from different seed examples? This is necessary because all these data points are used together as training data for training the Llama-based scorer. If they are not comparable, it might help to randomize the sets shown to ChatGPT for scoring.
- Relatedly, it would help to have humans also score data points in terms of complexity and quality and see if the scores correlate with
ChatGPT's judgments.

**Questions:**

- Is it possible for the diversity criterion to end up selecting outliers in the datasets? How can this issue be fixed?
- Do you need to use the representations from the Llama-13B model for computing distances as well? Can you use a different (smaller?) model for computing distances than the one you instruction-tune?
- DEITA-6K loses more to the random selection baseline than to the Vicuna model according to the human evaluation results in Table 4. This seems surprising. How does the random selection model perform compared to the Vicuna model in this case?

---

> ### Author Response · Authors · 2023-11-20
> **Response 1/2**
>
> Thanks for your helpful comments and we are glad that you acknowledged the strengths of our work! Below we will address your concerns point by point:
>
>
> ### Q1: Limited evaluation and more results on target benchmarks
>
> Thanks for the comments. We have added more evaluation results on the Open LLM Leaderboard benchmark [1] that consists of four tasks, ARC, HellaSwag, MMLU, and TruthfulQA.  Additionally, we have enriched our evaluations by training more pre-trained LLMs on our selected data, such as  LLaMA-2-13B and Mistral-7B besides LLaMA-1-13B in the original submission. We have summarized these results in the table of the general response, while the detailed results are in Table 5,6,7 of the paper. On the Open LLM Leaderboard, our approach outperforms the random baseline and other strong SOTA models across three different backbone models in terms of the average results. In the meanwhile, we would like to note that the performance on benchmarks such as MMLU is largely determined by the underlying base model, thus the gains after alignment tuning is typically not large, unless the training data is specifically crafted for the benchmark. This is why recent works on alignment put more emphasis on MT-bench and AlpacaEval [2].
>
> Notably, on MT-bench and AlpacaEval, our added results show that our method with the LLaMA-2-13B model surpasses Vicuna-13B-1.5 that trains on 10x more data, and LLaMA-2-13B Chat that has undergone extensive RLHF training. With Mistral-7B, our approach attains an MT-bench score of 7.29 with only 10k training data, marking a state-of-the-art performance on MT-bench *for all open-source SFT models at 7B and 13B sizes*. We also observe that, while the gains of the Mistral DEITA model on AlpacaEval are slightly lower than those on MT-bench, a deeper analysis (as depicted in Figure 4) indicates that the Mistral DEITA model excels particularly in mathematical and complex reasoning tasks, areas not primarily covered by AlpacaEval.
>
>
> ### Q2: GPT-4 evaluation bias
> This is a good point. As the reviewer cited,  [Wang et al. 2023](https://arxiv.org/abs/2306.04751) points out that win-rate scores given by GPT-4 correlate with length of responses. However, in section 5.4 of their paper, they also admit that human evaluation results largely correlate with the AlpacaFarm and benchmark-based evaluation due to the same trend in the comparisons among their Tulu-65B model and other models. On the other hand, several previous works have demonstrated that GPT4 annotator greatly correlates with human annotator [3, 4] with agreement ranging from 78%~85%. Based on these findings, while we recognize the limitations of GPT-4-based evaluations, we believe that GPT4-based evaluation is still valuable and able to provide useful insights of the models. In our experience, these evaluations often yield more reliable outcomes than those derived from average human annotators.
>
>
> ### Q3: Inter-annotator agreement of human evaluation
> Thanks for the advice! We would like to note that we tried hard to control the quality of human evaluation in the original submission – as described in Appendix D, we adopted MTurk in the beginning but abandoned their annotations due to low agreement scores among them. To justify our human evaluation, we follow LIMA [3] to randomly sample 50 generation pairs from our human evaluation dataset and compute the author-annotator agreement score – the resulting agreement score is 77%, which is reasonably high and aligns closely with the agreement levels reported in [3]. We have added the inter-annotator agreement score to Appendix D of the paper.

---

> ### Author Response · Authors · 2023-11-20
> **Response 2/2**
>
> ### Q4: Are these scores comparable across evolved sets from different seed examples?
> This is a good point. To answer this question, we conduct human evaluation on randomly selected 50 sample pairs from the ChatGPT complexity score pool – any example pair is not from the same evolved set. The authors then score each pair of examples from five categories: “A is significantly more complex than B”, “A is slightly more complex than B”, “neither is more complex”, etc. Then we compute the agreement score between the authors and ChatGPT – first, we transform the ChatGPT scores to pairwise judgment. We count it as “significantly better/worse” when the difference between ChatGPT scores is larger than 2, “slightly better/worse” for difference of 1, and tie in the case of 0. If ChatGPT has a high agreement score with the authors, that means the ChatGPT scores are comparable across evolved sets.
>
> In the “significantly better/worse” sample pairs, the ChatGPT scores are highly consistent with the authors, producing 83% agreement rate, which means the ChatGPT scores are comparable across evolved sets when the scores are not very close. In the “slightly better/worse” or tie categories, the authors also assign “slightly better/worse” or tie labels to 96% of such samples, suggesting high coarse-grained consistency. However, the fine-grained agreement on the three labels (slightly better/worse, tie) is only 43% – we find that it is hard even for humans to be consistent on comparing the slightly different complexity of instructions, especially when the two instructions are not relevant.
>
> We tried to repeat the above experiments for quality scores as well, but we found that it is really difficult for humans to compare the quality of two responses that correspond to two very different query instructions.
>
>
>
> ### Q5: Is it possible for the diversity criterion to end up selecting outliers in the datasets? How can this issue be fixed?
>
> We agree with the reviewer that the diversity criterion may select outliers in the dataset. . However, we think the outliers may not be always bad – being diverse and infrequent, the outliers may actually increase the diversity of the selected SFT dataset. Although outliers in a dataset are often considered as low-quality samples, our score-first data selection approach should have filtered out the low-quality samples in the first step.
>
> ### Q6: Can you use a different (smaller?) model for computing distances than the one you instruction-tune?
>
> We have performed ablation analysis on the embedding model in Appendix C of the original submission. Figure 4 shows that a smaller model, E5-Large-v2 (a SOTA sentence embedding model), is more sensitive to the threshold $\tau$, and its performance exhibits a significant decline compared to LLaMA..
>
> ### Q7: DEITA-6K loses more to the random selection baseline than to the Vicuna model according to the human evaluation
>
> We agree that this is a bit surprising, and we think this may be attributed to the simplicity of the LIMA test set, which fails to effectively distinguish between models that generally perform well. It is worth noting that the random baseline is not weak, as it is derived from a SOTA data pool. This situation highlights a broader challenge in human evaluation: for complex queries like those in MT-bench, human evaluation becomes difficult and costly, even for expert annotators, due to the complex examples which often include extensive mathematical and coding content. Conversely, with simpler queries like the LIMA test, human evaluators struggle to differentiate between strong models that excel in simple scenarios. To provide more clarity, we have included a detailed breakdown of the MT-bench results (based on GPT-4 evaluations) in Figure 4 of our paper. This breakdown clearly shows that the random baseline based on Mistral-7B lag significantly in areas such as coding, mathematics, and reasoning, aspects that are less prominent in the LIMA test and challenging for human evaluators to assess.
>
>
> [1] Beeching et al. Open LLM Leaderboard. https://huggingface.co/spaces/HuggingFaceH4/open_llm_leaderboard. 2023.
> [2] Tunstall et al. Zephyr: Direct Distillation of LM Alignment. https://arxiv.org/abs/2310.16944. 2023.
> [3] Zhou et al. LIMA: Less Is More for Alignment. https://arxiv.org/abs/2305.11206. 2023.
> [4] Zheng et al. Judging LLM-as-a-Judge with MT-Bench and Chatbot Arena. https://arxiv.org/abs/2306.05685. 2023.

---

> ### Author Response · Authors · 2023-11-22
> **A Kind Reminder for Reading the Response**
>
> Dear Reviewer NSy4,
>
> We have revised the paper and added many additional results to address your comments. Since the rebuttal period is closing very soon, can you please check the response to see whether it mitigates your concerns? We would greatly appreciate that!
>
> Thank you,
> The authors

---

### Author Response · Authors · 2023-11-20
**General Response to Reviewers and Revision Submitted**

We thank all the reviewers for their insightful comments and suggestions. We have revised the paper to address the reviewers’ concerns. Below we summarize the major revisions (the main revisions are marked with *blue* text in the pdf, we also made some minor layout changes to fit the page limit), while we reply to the comments of each reviewer separately.

The major revisions are:

1. We added the Open LLM Leaderboard [1] evaluation results (ARC, Hellaswag, MMLU,and TruthfulQA) besides MT-bench and Alpacaeval in the original submission (Reviewer NSy4, zdke).
2. We supplemented more human evaluation details on the inter-annotator consistency (Reviewer NSy4)
3. We added more baselines such as LIMA and Alpagasus (Reviewer zdke).
4. We reported additional strong experimental results of our approach based on LLaMA 2 and Mistral-7B models in addition to LLaMA 1 in the original submission. Such a comprehensive study marks our method as the **first work** on this research line to compete with SOTA SFT models with very limited training data selected automatically.  We hope these results address the concerns on limited evaluation from Reviewer NSy4, zdke, and the method’s contribution/significance over previous work from Reviewer RtKh.

We report a simplified version of some of the added results in the table below, and the detailed complete results are in Table 5,6,7 of the paper.



| Model | Data Size | MT-Bench | AlpacaEval | Open LLM Leaderboard (Avg.) |
|-------|-----------|----------|------------|--------------------------|
|**Open-sourced Models based on Mistral-7B**||||
| Mistral-7B-Instruct-v0.1                         | -- | 6.84     | 69.65 | 60.45     |
| Zephyr-7B-sft                                | 200K | 5.32     | 75.12       | 60.93      |
| Random-Select                        | 10K | 5.89     | 56.90        | 61.72      |
| $\text{DEITA}_{10K}$                         | 10K | **7.29**     | **80.59**       | **64.22**    |
|**Open-sourced Models based on LLaMA-2-13B**||||
| Vicuna-13B-v1.5                            | 125K  | 6.57     | 78.80       | 61.63      |
| Random-Select             |     10K     | 5.78    | 65.19        | 61.32     |
| $\text{DEITA}_{10K}$                 | 10K        | **6.79** | **81.09**   | **62.71**     |
|**Open-sourced Models based on LLaMA-1-13B**||||
| LIMA                                 | 1K       | 4.29     | 41.98       | 59.82     |
| WizardLM-13B          |        70K          | 6.35     | 75.31              | 58.96     |
| Vicuna-13B-v1.3           | 125K                  | 6.39     | **82.11**               | 60.01       |
| Random-Select                     | 10K | 6.03 | 71.52        | 60.14      |
| $\text{DEITA}_{10K}$        | 10K                 | **6.60**      | 78.01              | **64.27** |



[1] Beeching et al. Open LLM Leaderboard. https://huggingface.co/spaces/HuggingFaceH4/open_llm_leaderboard. 2023

---

### Meta-Review · Area_Chair_wpp7 · 2023-12-05

**Metareview:**

The paper study data selection for instruction tuning for LLMs. The proposed algorithm considers (1) complexity of query, (2) quality of the response, (3) diversity among the example set. They measure (1) and (2) for prompting ChatGPT, and (3) is enforced by greedy iterative addition (add only if they are far enough from its nearest example already chosen). They evaluate by fine-tuning Llama-13B model, showing solid gains over baselines. The paper is overall well-written and easy to follow, and the experiments are comprehensive.

**Justification For Why Not Higher Score:**

I am not sure about the impact of this paper, as the motivation is somewhat week. The instruction tuning part is not the most costly part of LLM training, not super convinced why data selection is very important here. It’s not even active learning, as labels are already all provided.

**Justification For Why Not Lower Score:**

The paper is clearly written and there’s no technical issues with the approach. The experiments are relatively comprehensive, and there's value in understanding what constitutes a good alignment dataset.

---

### Decision · Program_Chairs · 2024-01-16

Accept (poster)